# Kidney Function Worsening Is Linked to Parenteral-Nutrition-Dependent Survival in Palliative Care Patients

**DOI:** 10.3390/nu14040769

**Published:** 2022-02-11

**Authors:** Lea Kum, Alexander Friedrich, Markus Kieler, Elias Meyer, Petar Popov, Paul Kössler, Anna Kitta, Feroniki Adamidis, Raimund Oberle, Eva Katharina Masel, Matthias Unseld

**Affiliations:** 1Department of Medicine I, Division of Palliative Medicine, Medical University of Vienna, 1090 Vienna, Austria; lea.kum@meduniwien.ac.at (L.K.); alexander.friedrich@meduniwien.ac.at (A.F.); petar.popov@meduniwien.ac.at (P.P.); paul.koessler@protonmail.com (P.K.); anna.kitta@meduniwien.ac.at (A.K.); feroniki.adamidis@meduniwien.ac.at (F.A.); eva.masel@meduniwien.ac.at (E.K.M.); 2Institute for Vascular Biology, Centre for Physiology and Pharmacology, Medical University of Vienna, 1090 Vienna, Austria; markus.kieler@meduniwien.ac.at; 3Center for Medical Statistics, Informatics, and Intelligent Systems, Medical University of Vienna, 1090 Vienna, Austria; elias.meyer@meduniwien.ac.at; 4Institute of Medical Chemistry and Pathobiochemistry, Center for Pathobiochemistry and Genetics, Medical University of Vienna, 1090 Vienna, Austria; raimund.oberle@meduniwien.ac.at

**Keywords:** parenteral nutrition, kidney function parameters, serum creatinine, cancer, biomarkers, palliative care

## Abstract

Background. Parenteral nutrition (PN) is frequently administered in palliative care patients suffering from cachexia. The evidence regarding the use of PN in terminally ill patients is scarce. Routine laboratory parameters might help to decide whether to start or forgo PN, which could decrease overtreatment at the end of life. Kidney failure was frequently associated with survival. However, a relation between kidney function parameters and parenteral nutrition has not been observed thus far. The aim of this retrospective cohort study was to analyze kidney function parameters in palliative care patients under PN, as well as the relation between these parameters and overall survival. Methods. Patients who were admitted to the Department of Palliative Medicine at the Medical University of Vienna were screened for PN treatment. Whether kidney function parameters at baseline or their dynamics over the course of two weeks were associated with survival was assessed with descriptive and interferential statistics. Results. In total, 113 of 443 palliative care patients were administered parenteral nutrition for the first time. The overall survival (OS) for all patients with increased kidney function parameters at baseline was lower (creatinine: hazard ratio (HR) = 1.808, *p* < 0.001; urea: HR = 1.033, *p* < 0.001; uric acid HR = 1.055, *p* = 0.015). No significant increase in creatinine blood levels was observed in the first 2 weeks after the initiation of PN when compared to the non-PN group (*p* = 0.86). However, if creatinine blood levels increased within the PN group, lower overall survival was found (HR = 2.046, *p* = 0.007). Conclusion. Increased kidney function parameters, such as creatinine, urea and uric acid, might be used as negative prognostic markers in palliative care patients under PN. Moreover, an increase in creatinine during the administration of parenteral nutrition in the first 2 weeks is linked to worse outcomes. These findings may help future studies to establish objective markers for clinicians to determine whether to start or end PN in palliative cancer patients and decrease potential overtreatment at the end of life.

## 1. Introduction

Cancer-associated cachexia is very common in terminally ill patients. With a prevalence from 50 to 80% it affects nearly as many palliative care patients as pain [1,2,3,4]. However, unlike pain, treatment options remain limited [5,6]. This high prevalence makes cachexia to one of the most clinically relevant symptoms.

Tumor cachexia is the ongoing loss of skeletal muscles and adipose tissue with multifactorial causes such as metabolic changes due to the illness itself, antineoplastic treatment or medication for symptom management [7,8]. Loss of bodyweight leads to reduced physical function and weakness which affects patients in terms of their activities of daily living [9]. This can create a high burden of symptoms, as well as more and longer hospitalizations. All these factors contribute to poor quality of life [9,10,11,12].

There is no doubt that improved nutritional status is better for cachectic terminally ill patients and their quality of life, as well as their overall survival (OS) [12,13,14]. While this idea is backed by solid data for enteral nutrition, there is still a lack of evidence regarding parenteral nutrition (PN) [7,12,13,14]. Even though we know very little about the benefits of PN at the end of life, it is still a very common practice in terminally ill patients [15,16]. In palliative care units, PN is used in up to 10% of the patients, whereas in other medical specialties, artificial nutrition is administered in up to 53% of patients at the end of life [14]. Tobberup et al. claim that PN improves nutritional state but has no effects on the patient’s quality of life. Moreover, they state that PN has no benefits in terms of OS compared to intravenous fluids [17]. With the positive effect of improving nutritional status comes the negative effect of risking infections [18,19,20].

Given this controversial topic, there is a need for clear parameters and guidelines regarding when to start and when to stop administrating PN to patients at the end of life. The evidence is scarce, but it has been suggested that initiation of PN is indicated when life expectancy is longer than 1 to 3 months and the patient is expected to die from cachexia rather than tumor spread [1,6,20,21,22].

The necessity for an objective score that helps in predicting survival to improve decision making at the end of life was also the main motivation for Chen et al. to develop an objective palliative prognostic score (OPPS) for patients with advanced cancer. With six variables including serum creatinine that could relatively accurately predict that death would occur within seven days, this score might be helpful to decide whether to prolong or forgo PN treatment. However, Chen et al. aimed to create a score which helps in decision making at the end of life in general but did not specify on PN treatment [23].

One benefit of the above-mentioned objective score may be that it does not rely on subjective variables such as patients’ symptoms or condition and physicians’ experiences. These variables are used in established and validated prognostic scores, such as the Palliative Prognostic Index [24], Palliative Prognostic (PaP) Score [25] and Prognosis in Palliative Care Study (PiPS) [26]. Most of these commonly used scores aim to predict short time survival over the next days or weeks. The authors suggest the mentioned scores to be helpful in deciding whether to give antineoplastic treatment [25,27] or medication [24]. Although nutrition is an essential part of palliative treatment none of the above mentioned scores focuses on the decision whether to start or forgo PN treatment at the end of life. 

When it comes to predicting survival, most authors and clinicians came to the same conclusion. The most accurate way to determine a prognosis was a combination of an objective prognostic score and the estimate of a multiprofessional team [23,26,27]. As a common objective marker impaired renal function is commonly found [23,28]. In contrast to that, Aung et al. suggested believed that a low serum creatinine would be predictive of shorter survival because it reflects the level of cachexia in a patient. They hypothesized that since creatinine levels are influenced by muscle mass and cachectic patients do have a worse prognosis there has to be a relation. They retrospectively considered the data of 83 palliative care patients over a period of 2 months and found their hypothesis to be wrong. Survival in patients with creatinine less than 0.91 mg/dL was better than with creatinine over 0.91 mg/dL (*p* = 0.035) [29].

Given the current level of evidence, it is difficult to determine whether initiating PN benefits terminally ill patients. Thus, it would be helpful to have defined prognostic markers that predict survival. This study intends to analyze the relation between kidney function parameters and overall survival under parenteral nutrition. This should support future decision making and minimize over treatment at the end of life.

## 2. Materials and Methods

### 2.1. Study Design and Population

This retrospective cohort study used the laboratory parameters of patients admitted to the palliative care unit (PCU) of the Medical University of Vienna over a time period of two years. We did use all registered data from January 2016 until January 2019. Out of 443 patients, only two patients were excluded due to their missing PN status. The patient data were entered anonymously from digital sources. Data were checked by two collaborators independently. The data file was only administered and stored on password-protected computers in order to guarantee data protection of each patient. The present study was conducted in accordance with the Declaration of Helsinki (World Medical Association, 2013). Ethical approval was gained by the Ethics Committee of the Medical University of Vienna (EK Nr: 2185/2018).

### 2.2. Setting and Data Collection

We collected data of 443 patients from the electronic database of the Medical University of Vienna. Since this is a retrospective study, we used registered data and laboratory values which were routinely collected within the admission. Further data from routine blood sampling at the Palliative Care ward were used. Baseline data (age, sex and body mass index (BMI)) were used from the day of admission. Laboratory parameters were collected at two time points. First, data were collected on the day of admission (T0). The second dataset was obtained in the second week after the initiation of PN or after admission for the No-PN Group (T1). This was completed in order to identify the dynamics of the laboratory parameters under PN administration. Due to the retrospective design of this study we had to use registered data. Therefore, the time between T0 and T1 was approximately 10 days (±3 days).

### 2.3. PN Regimens

Due to the individual needs of each palliative care patient, the decision regarding whether to administer PN is made by the dietologist of the division. The PN used on the unit was NuTRIflex^®^ Omega special (625 mL bag with 740 calories, 35 g of protein, 90 g of carbohydrates and 25 g of fat; B. Braun Melsungen AG, Germany, 2014), combined with a balanced mixture of Soluvit (vitamins: b1, b6, b12, c, nicotinamide, pantothenic acid, biotin and folic acid; Fresenius Kabi Austria GmbH, Austria, 2013), Vitalipid (contains vitamins: a, d2, e and k1; Baxter Deutschland GmbH, Germany, 2015) and Trace (contains trace elements: Fluorine (F), Iodine (I), Molybdenum (Mo), Iron (Fe), Copper (Cu), Manganese (Mn), Selenium (Se) and Zinc (Zn) as well as electrolytes; Fresenius Kabi Austria GmbH, Austria, 2018). It is usually administered overnight with a target energy intake of 1475 ccal/d.

### 2.4. Statistical Analysis

Continuous variables were summarized as means/medians and standard deviation/quantiles as appropriate. Categorical variables were summarized using absolute and relative frequencies. In order to assess whether the population of patients receiving PN differs from that receiving no PN in terms of blood levels and baseline values (age, sex and BMI), univariate logistic regression models were used. Changes in blood levels from T0 to T1 were analyzed with a paired t-test. To assess the association between the parameters of interest and OS, a univariate Cox-regression was conducted, separately for the PN and No-PN group, using OS as the dependent variable and the parameter of interest as the independent variables. Furthermore, to determine the influence of PN on the overall survival, a multivariable Cox-Regression adjusting for the confounders age, gender and BMI was fitted. *p*-Values < 0.05 were considered statistically significant. As *p*-Values serve only descriptive purposes, no multiplicity corrections were applied. A hazard ratio (HR) > 1 indicated a greater risk of death, while an HR < 1 indicated a lower risk of death or better survival.

## 3. Results

### 3.1. Patients’ Characteristics

This study was conducted at the PCU of the Medical University of Vienna. As shown in Table 1, 245 (55%) female and 198 (45%) male patients were included, of whom 26% (113 patients) received PN during their hospitalization. The mean age of patients receiving PN was 60 years (No-PN 65 years) and their mean BMI was 20.02 kg/m^2^ (±3.36 kg/m^2^), as compared to 24.36 kg/m^2^ (±15.02 kg/m^2^) in the No-PN Group. Forty-one percent of the patients who received PN suffered from cancer originating in the gastrointestinal tract (2% in the No-PN-group). In 88%, metastatic disease was present (85% in the No-PN Group), while the other 12% (No-PN: 15%) suffered from locally advanced cancer stages.

### 3.2. Comparison of Patients’ Characteristics and Overall Survival

At baseline, the following patient parameters were associated with receiving PN: younger age (OR = 0.97; *p* < 0.001), lower BMI (OR = 0.82; *p* < 0.001), lower serum creatinine (OR = 0.7; *p* = 0.032) and lower urea (OR = 0.99; *p* = 0.04)—as shown in Table 2. In a multivariable regression model, the OS—measured from the day of admission to the PCU—did not differ between the PN and No-PN group (*p* = 0.673). There was also no significant association found between gender (female, HR = 0.832; *p* = 0.087) or BMI (HR = 1.014; *p* = 0.171) and OS. Only age was significantly associated with OS, with a higher age indicating lower OS (HR = 0.991; *p* = 0.041).

### 3.3. Clinical Outcome and Association with Laboratory Variables at Baseline

In both groups (PN/no-PN) patients with higher levels of creatinine, measured on the day of admission to the PCU, had lower OS (HR = 1.808; *p* < 0.001/HR = 1.179; *p* = 0.002). This association could also be found with urea (HR = 1.033; *p* < 0.001/HR = 1.016; *p* < 0.001) and uric acid levels (HR = 1.055; *p* = 0.015/HR = 1.09; *p* < 0.001). Results are shown in Table 3.

### 3.4. Dynamics of Kidney Function Parameters and Their Association with Clinical Outcomes

In both groups serum creatinine levels did not change significantly from T0 (day of admission) to T1(second week after admission). However, a one unit increase in serum creatinine from T0 to T1 was associated with a significantly lower OS in patients receiving PN (HR = 2.036; *p* = 0.007).

A similar but less pronounced effect was observed in patients who did not receive PN (HR = 1.261; *p* = 0.054). Results are shown in Table 4.

## 4. Discussion

With nearly the same prevalence as pain, tumor-cachexia is clinically relevant and affects up to 80% of palliative patients [2,3,4,5]. The loss of skeletal muscle and adipocytes has multifactorial causes and may be caused by chronical illness itself or tumor treatment [7,8]. Tumor cachexia is associated with not only shorter survival but also decreased quality of life, which is the major concern in palliative care [9,10,11,12]. Improving the nutritional status of cachectic patients is, without doubt better for their wellbeing. Data on enteral artificial nutrition show its benefits, while for parenteral nutrition, there still is a lack of evidence. Nonetheless, parenteral nutrition treatment is a frequently used method in palliative patients. As the data shows, up to 53% of patients are treated with PN at the end of life [6,12,13,14].

Even though PN is commonly used at the end-of-life evidence supporting this practice is very scarce [6]. With our findings we hope to support future studies to find an objective score that will help in the decision-making process for clinicians. We found that serum creatinine did not significantly increase under PN treatment. However, for those patients who showed an increase in serum creatinine during the administration of PN, the OS was lower. The same effect did show in the No-PN group, but it was less pronounced and not statistically significant.

We think our findings are of importance for the palliative medical field since it is the first study to suggest a link between kidney function parameters and parenteral-nutrition-dependent survival. There is little to no evidence in combining a prognostic score with the PN dependent survival. There is one study by Llop-Talaveron et al. combining an inflammation based prognostic score with the clinical outcome of patients under PN treatment. Even though the design and patient cohort is different to ours this study showed the impact of inflammatory parameters on the survival of patients with PN. They concluded that the systematic use of prognostic scores before the initiation of PN might help to figure out whether a patients will profit from PN or not [30]. 

Furthermore, we could observe that patients who had higher blood levels of creatinine, urea and uric acid at baseline had lower OS. The latest version of the ESMO Clinical Practice Guidelines on cancer cachexia in adult patients [6] suggests initiation of PN only if the life expectancy is longer than 3 to 6 months. Therefore, our findings may be useful in the decision-making process and may decrease overtreatment at the end of life [6].

Even though impaired kidney function has been associated with a shorter OS in earlier studies [28,31] our findings are important because creatinine is not yet a common prognostic factor in the palliative care field. Tanaka et al. did include urea in their prognostic laboratory score among with seven other routine laboratory parameters. With their newly developed score (C-reactive protein ≥ 6.8 mg/dL, aspartate aminotransferase ≥ 43 U/L, blood urea nitrogen ≥ 22 mg/dL, white blood cell count ≥ 10.9 ×10^3^/μL, eosinophil percentage ≤ 0.4%, neutrophil-to-lymphocyte ratio ≥ 12.0, red cell distribution width ≥ 16.8 and platelet count ≤ 168 × 10^3^/μL) they could relatively accurately predict 14-day mortality in terminally ill patients with a malignant lung disease [31]. In addition to urea as the only laboratory marker Chiang et al. did include patient’s cognitive status, edema, ECOG (Eastern Co-operative of Oncology Group) performance status and respiratory rate in their prognostic 7-day survival formula for patients with terminal cancer [28].

Our findings might also be of interest since newly developed instruments, such as the CONUT (Controlling Nutritional Status) Score [32], mainly focus on inflammatory parameters to predict survival rather than impaired kidney function. With this score the immune–nutritional status and furthermore the prognosis in some cancers were to be predicted [31]. When it comes to decision making in respect of the prognosis the modified Glasgow Prognostic Score (mGPS) [33] is one of the most validated tools. The mGPS was designed to predict survival in cancer patients using only C-reactive protein and albumin [33]. Considering our findings the inclusion of kidney function parameters in prognostic scores should not be missed. Especially when it comes to decision making concerning the initiation of PN treatment.

Decision making at the end of life can be difficult for medical professionals in general. This is why there have been many approaches to implement guidelines for example to consider the necessity of invasive procedures such as parenteral nutrition [34]. Predicting the patient’s survival is only one of many factors that has to be considered before deciding whether to start or forgo any procedure. Predictive prognostic scores can be helpful when deciding whether to start or forgo medical treatment of any kind. The level of evidence for PN in terminally ill patients is very low [6]. Our findings may support further work and research concerning this important topic at the end of life.

This study has several limitations that warrant discussion. It was a non-randomized and retrospective analysis of a single center registry. The study cohort lacks homogeneity between the described groups. Patients suffered from a wide range of different cancer types and treatment. In addition, mean enteral feeding time was not available for analysis, further complicating subject comparability. Due to the cross-sectional design, data from only two timepoints were available thus, the dynamic of the laboratory parameters might not be displayed properly. Due to the descriptive purpose of this study, we included all patients who have been administered at the PCU to avoid a possible selection bias. Additionally, due to the retrospective design, we could not assess further influencing factors such as information on quality of life improvements. 

Yet, this is the first study linking kidney function worsening to parenteral-nutrition-dependent survival in palliative care patients. Future research on this issue is warranted to understand the complex metabolic alterations and its impact on patient survival and quality of life.

## 5. Conclusions

All together this study suggests increased kidney function parameters at baseline as negative prognostic markers in palliative care patients in general as well as under PN. Moreover, a kidney function worsening during the administration of parenteral nutrition in the first 2 weeks is linked to worse outcomes. These findings may help future studies to find an objective score for clinicians to determine whether to start or end PN in palliative cancer patients and decreases potential overtreatment at the end of life.

## Figures and Tables

**Table 1 nutrients-14-00769-t001:** Patients’ characteristics.

		Total	No-PN Group	PN Group
(*n*, %)		443 (100)	330 (74)	113 (26)
Female (*n*, %)		245 (55)	183 (56)	62 (55)
Male (*n*, %)		198 (45)	147 (44)	51 (45)
Mean Age (SD, years)		63.6 (12.5)	64.71 (12.08)	60.12 (12.96)
Mean BMI (SD)		23 (5.3)	24.63 (15.02)	20.02 (3.36)
<18 (*n*, %)		92 (21)	47 (14)	45 (40)
18–25 (*n*, %)		228 (51)	173 (52)	55 (49)
25–30 (*n*, %)		43 (10)	37 (12)	6 (5)
>30 (*n*, %)		36 (8)	34 (10)	2 (2)
n.d. (*n*, %)		44 (10)	39 (12)	5 (4)
Mean Creatinine (SD, mg/dL)		1.1 (0.9)	1.1 (0.9)	0.9 (0.7)
Mean Urea (SD, mg/dL)		24.3 (20)	25.4 (21.5)	20.9 (14.1)
Mean Uric Acid (SD, mg/dL)		6.1 (3.7)	5.8 (3.6)	6.2 (3.7)
Tumor Origin (*n*, %)				
Gastrointestinal Tumor		124 (28)	78 (24)	46 (41)
HCC ^1^/CCC ^2^		16 (4)	11 (3)	5 (4)
Lung		85 (19)	74 (22)	11 (10)
Breast		53 (12)	47 (14)	6 (5)
ENT ^3^		24 (5)	17 (5)	7 (6)
Reproductive organs		27 (6)	16 (5)	11 (10)
RCC ^4^/Urothelial		17 (4)	15 (5)	2 (2)
Sarcoma		25 (6)	18 (6)	7 (6)
Blood		23 (5)	17 (5)	6 (5)
NET ^5^		7 (1)	4 (1)	3 (3)
Brain		13 (3)	10 (3)	3 (3)
Other		29 (7)	23 (7)	6 (5)
Metastasis	Yes	380 (86)	280 (85)	100 (88)
	No	63 (14)	50 (15)	13 (12)

^1^ Hepatocellular Carcinoma, ^2^ Cholangiocellular Carcinoma, ^3^ Ear Nose Throat Tumor, ^4^ Renal Cell Carcinoma, ^5^ Neuroendocrine Tumor, SD = Standard Deviation.

**Table 2 nutrients-14-00769-t002:** Baseline Variables and their association with the initiation of PN.

	OR (CI)	*p*-Value
** *Sex* **	0.98 (0.64; 1.5)	0.915
** *Age* **	0.97 (0.95; 0.99)	<0.001
** *BMI* **	0.82 (0.77; 0.87)	<0.001
** *Creatinine* **	0.7 (0.51; 0.97)	0.032
** *Uric Acid* **	0.97 (0.91; 1.03)	0.333
** *Urea* **	0.99 (0.97; 1)	0.04

OR = Odds Ratio, CI = 95% Confidence Interval of the OR.

**Table 3 nutrients-14-00769-t003:** Laboratory variables at baseline and their association with overall survival.

		HR (CI)	*p*-Value
**Female**	PN Group	0.856 (0.579; 1.267)	0.437
	No-PN Group	0.855 (0.675; 1.084)	0.196
**Age**	PN Group	0.996 (0.982; 1.011)	0.593
	No-PN Group	0.994 (0.985; 1.004)	0.228
**BMI**	PN Group	1.052 (0.997; 1.11)	0.063
	No-PN Group	1.01 (0.988; 1.033)	0.364
**Creatinine**	PN Group	1.808 (1.352; 2.419)	<0.001
	No-PN Group	1.179 (1.061; 1.311)	0.002
**Urea**	PN Group	1.033 (1.018; 1.048)	<0.001
	No-PN Group	1.016 (1.011; 1.02)	<0.001
**Uric Acid**	PN Group	1.055 (1.01; 1.102)	0.015
	No-PN Group	1.09 (1.057; 1.124)	<0.001

HR = Hazard Ratio, CI = 95% Confidence Interval.

**Table 4 nutrients-14-00769-t004:** Association of change in serum creatinine values from T0 to T1 and overall survival.

Diffenrences in Creatinine	HR (CI)	*p*-Value
PN	2.036 (1.21; 3.426)	0.007
No-PN	1.261 (0.996; 1.596)	0.054

HR = Hazard Ratio; CI = 95% Confidence Interval.

## Data Availability

Data will be made available upon request to Matthias Unseld, M.D. Ph.D., at matthias.unseld@meduniwien.ac.at.

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
