# Peer review of "Kidney Function Worsening Is Linked to Parenteral-Nutrition-Dependent Survival in Palliative Care Patients"

_nutrients, 2022, doi:10.3390/nu14040769_

Round 1

Reviewer 1 Report

The study "Kidney-Function Parameters Predict Parenteral-Nutrition-Dependent Survival in Palliative Care Patients" aims to analyze kidney function parameters in palliative care patients under parenteral nutrition, as well as the correlation between these parameters and overall survival.

From a general point of view, the manuscript aims to facilitate decision-making and determine whether to initiate or terminate PN in palliative cancer patients and to reduce potential overtreatment at the end of life.

The authors have tried to carry out a study, but there are some aspects that should be considered:

Major comments:

  1. The abstract does not identify the type of study that has been conducted.
  2. The introduction should incorporate more evidence through the use of bibliographical references. There are long sentences that do not incorporate references. If possible, the statements or sentences should be referenced, and avoid putting the citation at the end of a paragraph.
  3. There are paragraphs in the introduction that are presented in isolation (e.g. lines 55-58). This paragraph should be linked to the text in an insightful way. It is an isolated paragraph, not specific to the topic and objectives. Please introduce more evidence on the topic under study, an example of evidence is the paragraph between lines 88 and 93.
  4. On line 96 the term "correlation" is used. Please, since the term "correlation" is used in the introduction section, it is preferable to use the term "relation". In the introduction section, we should not yet know whether or not "correlation" exists, so the aim of the study should be to analyze the existence of "relation".
  5. The "Material and methods" section needs brief considerations:
    • Please specify in detail the type of study (section 2.1), the period in which the data were collected, as well as the date of the original data - e.g.: "...data were collected from June 2021 to August 2021. The data selection period was from January 2018 to December 2020". It should state and include the protocol number of the study, as well as the type of committee that reviewed the study. It should also explain in detail how the data were shielded and anonymized.
    • Please determine in section 2.2. the specific inclusion and exclusion criteria that were used. Determine the ranges of age, education, etc... Indicate the type of sampling used. The sentence "First, data was collected on the day of admission (T0). For the PN group this was before the initiation of PN" is unclear and may mislead the reader. Baseline should have been measured in all groups in a similar way, if not please specify the measurement criteria used in all groups. The sentence "The second dataset was obtained two weeks after the initiation of PN or after admission for the No-PN Group" should be clarified. The range between the dates of measurement in both groups should be specified. In case of non-homogeneity (as this is a retrospective study), please be explicit and specify the reasons for this.
    • The mean enteral nutrition time should be specified, together with the standard deviation (section 2.3).
    • If there were 2 measurements, please use T0 and T1 (T2 would result in the existence of a partial measurement in the interval, which would be T1). If there was a partial measurement in the middle of the interval, then you must specify which statistical test was used (e.g. ANOVA, etc...).
  6. The "Results" section needs to be comprehensively restructured. The results are not clearly categorized and are vague to read.
    1. On line 135, is it median or mean?
    2. The data expressed in lines 135-141 may be expressed in a standardized form. Please use the following model: "...and BMI (20.02 ± 3.36 kg/m2)".
    3. Please use 3 columns to describe Table 1. One column should include the descriptive parameters of the total of all variables, the second column should include the parameters of the [No-PN Group], the third column should include the parameters of the [PN Group]. All variables should present data according to the specified breakdown. Do not submit aggregate data (e.g. sex), which are not included in each of the specified groups (Total, No-PN Group, PN Group).
    4. Please standardize the notation. In table 2 the OR statistic is reported as OR, in line 152 it is reported as HR.... Standardize the decimal score...
    5. Table 2 states "This is a table. Tables should be placed in the main text near to the first time they are cited". Please remove this text and specify the name of the table. Indicate in the table footer the meaning of CL and OR. Please enter the 95% confidence interval of the OR values in brackets after their value.
    6. In section 3.3 the description may be unclear. Please specify what you are measuring. If you are measuring OR, why do you specify "correlation"?
    7. Table 3 states "This is a table. Tables should be placed in the main text near to the first time they are cited". Please delete this text. Point out in the table footnote the meaning of CI and OR. Please enter the 95% confidence interval of the OR values in brackets after your value. Why are the OR values of the variable [Uric Acid] not presented?
    8. In line 165 "T2" appears when it should be T1, as mentioned above in the methodology section.
    9. Table 4 states "This is a table. Tables should be placed in the main text near the first time they are cited". Please remove that text. Insert the description of the table. This table needs to be restructured as 2 different analyses are presented which may lead to misinterpretation.
    10. The authors have run a logistic regression model. Please specify whether any fitting of the model has been performed. Could you please provide information on the Nagelkerke statistic?
  7. The DISCUSSION section needs to be restructured:
    1. In line 183 do not use the word "paper"; please use "study".
    2. On line 185 the word "correlated" is used. Are you sure about that? In any case, the analysis has been a hazard?
    3. The paragraph between lines 185-188 should be linked to the text in an insightful way. Please do not duplicate text that is already specified in previous paragraphs. Delete this text, as it is redundant here.
    4. Please try to introduce evidence that confronts or is similar to your findings. Do not repeat text you have used in the introduction. Rephrase the sentences and connect them to each other.
    5. The discussion of the study is incomplete and poor. No reference is made to similar studies that may share or refute the data obtained in the study.
  8. The REFERENCES section needs to be thoroughly revised.
    1. Please check that the references are in MDPI format, as there are errors in the referencing styles.

Minor comments:

  1. Please use the uniform p-Value style throughout the text (e.g. p-Value).
  2. Please specify values with decimals in text and tables (unify criteria and styles).

I encourage you to continue to carry out such studies. The monitoring of the variables used in the study can be of great help in planning end-of-life care. Work along these lines will contribute significantly to decision making, provide solid scientific support for future implementations, and effectively and efficiently optimize the use of employee resources in healthcare institutions.

Reviewer 2 Report

This study from Kum and colleagues aims to analyze the association between kidney function and survival in palliative care patients on parenteral nutrition. 

Major comments:

1. Unfortunately, the methods and results sections are sometimes weak and difficult to understand. In my view, given the aim of the study, too much attention is oriented to no-PN vs PN patients and to baseline characteristics. The summarized statistical analysis are difficult to interpret, particularly for continuous variables. This is also true because of the absence of Tables' titles or captions (Table 2 to 4), the reporting of the statistical analysis section (which I believe does not follow the same order of the results presentation), and some tables' "mislabelling" (possibly OR instead of HR in Table 3 and 4). 

2. The final message of the study is that of an association between kidney function worsening and shorter survival in those receiving PN. However, serum creatinine increase is defined by only two measures (which is not ideal per se and streches the concept of "dynamic"), at two time points not clearly defined in time (T0 to "two weeks after PN initiation"). Table 4a shows no difference between T0-T2 serum creatinine means so that it is even more difficult for the reader to understand the extent of the increase the Author propose to be clinically significant. Maybe graphs or descriptive data could help clarify these points. 

3. In the conclusion section the Authors suggest kidney function parameters as negative prognostic markers in palliative care patients under PN. This seems to me stretched from the presented results which may show a statistical association, in univariate analysis, but no predictive values as per definition. Moreover, based on the results from Table 3, the same association should be described for no-PN patients. For these reasons I suggest also to rephrase the manuscript title.

Minor comments:

a. In the introduction section lines 80-87, the Authors should address if any of the scores they mention indicates if and when to benefit from PN.

b. Occasionally values are missing from the tables (ie, Table 3 uric acid's OR) or mistakenly reported.

c. I would suggest to avoid the expression "increased kidney function parameters" and similar in favor of "kidney function worsening"

d. In the conclusion section, lines 220-1, the sentence should include "at baseline" for clarity.

e. Even if the topic is somehow niche, the cited reference are not extremely current with only 11/23 from 2016 or after. 

Round 2

Reviewer 1 Report

I congratulate them for taking into consideration the suggestions made in the first revision. The authors have mostly addressed the suggestions made and the manuscript has increased in quality and readability considerably. However, there are still some considerations that should be taken into account and addressed by the authors. The most important of these are outlined below:

Major comments:

  1. On line 28 of page 1, please enter the p-value at the end of the sentence. Use the standardized form of expression of "p". In addition, it would be advisable to enter the mean values obtained in each group.
  2. The paragraph on lines 68 to 79 on page 2 is referenced several times with the same quotation at the end of each sentence. Please review this paragraph and try to adapt it accordingly.
  3. Thank you for expanding the sentence on the selection criteria (inclusion and exclusion) of the sample. However, using "all patient records" can be imprecise. Were patients not excluded according to age, pathology, etc.? Were the records not selected at standardized intervals? - The use of this type of sample selection can be quite imprecise and the results unclear. If "all" registers were used, without any criteria, this is a clear threat to validity, which should be clearly reflected in the corresponding "study limitations" section.
  4. If the authors have not been able to address the issue of "mean enteral nutrition time" specified in the previous review, it should be considered as a clear threat to the validity of the study and should be included in the corresponding "study limitations" section.
  5. In the "conclusions" section, the authors specify "These findings may help clinicians to determine whether to start or end PN in palliative cancer patients and decreases potential overtreatment at the end of life". Due to the type of study and methodology used, as well as the limitations of the study, the sentence should be rephrased. Indeed, the retrospective study performed would help future studies with robust designs that could correctly describe whether increased renal function parameters are a negative prognostic marker in palliative care patients. Do not forget to rephrase the sentence in the Abstract and Conclusions sections.

Minor comments:

  1. Please use the uniform p-Value style throughout the text (e.g. p-Value).
  2. References do not appear to be properly annotated. Normally, the reference is annotated after the dot. Please check it strongly.
  3. In line 159, "kg/m2" appears again, please use the superscript "kg/m2".
  4. Thank you for specifying abbreviations appropriately; however, in Table 2, 3 and 4, the authors use "CL" when they should use "CI". Please specify the confidence interval in the tables accordingly (e.g. [0.64 - 1.50]).

Again, I congratulate you on the work you have done. Good luck!

Reviewer 2 Report

The Authors thoroughly addressed all comments and significantly improved the quality of the manuscript. I am not an expert in statistical analysis, but two questions keep coming to my mind and I think the Authors should try (if possible) to clarify (methods and results) about the main finding of the study (ie, Table 4):

a) how is the worsening in serum creatinine defined (threshold? delta?)?

b) how many patients (n, %) showed the worsening in the two groups?

I apologize if these data are not available due to the type of statistical test and for not being exhaustive in my first report.
